# Endophyte Community Changes in the Seeds of Eight Plant Species following Inoculation with a Multi-Endophytic Bacterial Consortium and an Individual *Sphingomonas wittichii* Strain Obtained from *Noccaea caerulescens*

**DOI:** 10.3390/plants12203660

**Published:** 2023-10-23

**Authors:** Tori Langill, Małgorzata Wójcik, Jaco Vangronsveld, Sofie Thijs

**Affiliations:** 1Environmental Biology, Centre for Environmental Sciences, Hasselt University, Agoralaan Building D, 3590 Diepenbeek, Belgiumjaco.vangronsveld@uhasselt.be (J.V.);; 2Department of Plant Physiology and Biophysics, Institute of Biological Sciences, Maria Curie-Skłodowska University, 20-033 Lublin, Poland

**Keywords:** community profiling, growth promotion, hyperaccumulator, *Noccaea caerulescens*, seed endophytes, seed inoculation

## Abstract

*Noccaea caerulescens*, a hyperaccumulator plant species known for its metal tolerance and accumulation abilities, harbours a microbiome of interest within its seed. These seed-associated bacteria, often referred to as seed endophytes, play a unique role in seed germination and plant growth and health. This work aimed to address how inoculating seeds of eight different plant species—*Medicago sativa* (alfalfa), *Zea mays* (corn), *Raphanus sativus* (radish), *Helianthus annus* (sunflower), *Cucurbita pepo* subsp. *pepo* (squash), *Beta vulgaris* subsp. *cicla* (rainbow chard), *Arabidopsis thaliana* (thale cress), and *Noccaea caerulescens* (penny cress)—with a bacterial consortium made from the seed endophytes of *N. caerulescens* would affect the seed microbiome of each test plant species, as well as inoculation with a strain of the bacterium *Sphingomonas wittichii*, which was previously isolated from seeds of *N. caerulescens*. Additionally, we aimed to offer preliminary plant tests in order to determine the best seed treatment plan for future research. The results showed that inoculation with the bacterial consortium held the most potential for increasing plant size (*p* < 0.001) and increasing germination rate (*p* < 0.05). The plant that responded best to inoculation was *N. caerulescens* (penny cress), likely because the microbes being introduced into the seed were not foreign. This paper also offers the first insight into the seed endophytes of *Beta vulgaris* subsp. *cicla*, highlighting an abundance of Proteobacteria, Firmicutes, and Actinobacteriota.

## 1. Introduction

Plants are not solitary organisms but rather host a diverse array of microorganisms that reside within various plant tissues. Among these microbial communities, seed endophytes have emerged as a significant and intriguing group of microorganisms that inhabit the internal tissues of seeds [1]. Seed endophytes, encompassing bacteria, fungi, and other microbial taxa, have garnered considerable attention in recent years due to their potential contributions to plant health, growth promotion, and ecological adaptations [2,3,4]. Investigating seed endophytes and their interactions with plants provides a fascinating avenue to uncover the hidden microcosm within seeds and unravel the intricate relationships between plants and microorganisms.

Seed endophytes have been found in a wide range of plant species across diverse ecosystems, indicating their ubiquitous presence in nature. These microorganisms colonize the embryo, endosperm, or seed coat and can be vertically transmitted from one generation to the next [1,5]. This vertical transmission enables seed endophytes to play a critical role in shaping the plant microbiome from early developmental stages, and means that the selected seed microbiome is preserved across generations [6].

Understanding the functional significance of seed endophytes has gained prominence due to their potential contributions to plant health and productivity. Numerous studies have revealed the ability of seed endophytes to enhance nutrient acquisition [7], improve stress tolerance [4,8], and promote plant growth [9,10]. These beneficial effects are attributed to various mechanisms, including nitrogen fixation, phosphate solubilization, production of plant-growth-promoting substances, and modulation of the plant’s immune system. By harnessing the potential of seed endophytes, it may be possible to develop sustainable agricultural practices that minimize the reliance on chemical inputs while simultaneously increasing crop yields and resilience [11].

This paper focusses primarily on how inoculating seeds of crop species with seed endophytes from a resilient plant species can affect the crops’ internal seed microbiome, as well as address the question of whether it is better to inoculate seeds with a single plant-growth-promoting endophyte, or a consortium derived from seeds of the resilient plant in question.

*Noccaea caerulescens*, a hyperaccumulator plant species, has garnered significant importance in the field of phytoremediation and ecological research, making it an ideal candidate for our resilient plant. This metallophyte exhibits an exceptional tolerance to and accumulation of heavy metals, particularly zinc, cadmium, and nickel, in its aboveground tissues [10]. Still, this plant is not only found in polluted areas but also has a strong presence in dry hillside meadows, gardens, bare ground, and forests [12]. The common name is penny cress due to the purse shape of the plant’s seed pods (Figure 1).

The samples of *N. caerulescens* used for this research originated from the historic mining and smelting site in Plombières, Belgium. Because of these activities, the soil is heavily polluted with lead, cadmium, and zinc [13]. This important area is covered with smelter slags. This site was chosen to ensure the penny cress plants producing our seeds were grown in stressful conditions.

Previous research has shown *Sphingomonas wittichii* to be a seed endophyte present in *N. caerulescens* [10]. This particular endophyte has piqued the interest of the scientific community due to its ability to degrade environmental pollutants, such as dioxins [14,15,16]. In addition to the outstanding degradation properties that *S. wittichii* possesses, it also previously showed promise in increasing plant biomass [10]. Because it was one of the few endophytes isolated from *N. caerulescens* that was able to be grown in the lab and held promise for seed inoculation, it was chosen as the isolate to test for the seed inoculation potential of crops.

The chosen crops of interest were as follows: *Medicago sativa* (alfalfa), *Zea mays* (corn), *Raphanus sativus* (radish), *Helianthus annus* (sunflower), *Cucurbita pepo* subsp. *pepo* (squash), *Beta vulgaris* subsp. *cicla* (rainbow chard), and *Arabidopsis thaliana* (thale cress). Inoculation was also applied to the seeds of *N. caerulescens* in order to offer greater insight into what effect seed inoculation can have on the seed microbiome of plant seeds.

This research aims to address the community changes within the seed endophytic community of the aforementioned crops following inoculation with a bacterial isolate (*S. wittichii*) as well as inoculation with a bacterial consortium made from the endophytes present in the *N. caerulescens* seed. Additionally, a follow-up in planta experiment was performed using seeds of *N. caerulescens*, *A. thaliana*, and *M. sativa* in order to offer insight into whether inoculation with an isolate is indeed better than inoculation with a relatively uncontrolled consortium. It is hypothesized that using a consortium will have the same effect as an isolate. If this hypothesis is supported, it could be a step towards making a more cost-effective, easier-to-maintain approach for industries who may be interested in seed inoculation with endophytes. Additionally, it is hypothesized that the seed inoculation method mentioned in this paper will have an obvious and significant effect on the internal microbiome of all seeds, regardless of shape, seed coat thickness, and size.

This paper also brings awareness to the initial endophytic communities residing in the above-mentioned crops through Illumina sequencing, which helps further the fundamental research progressing in each area of study, which would use them as a test species.

## 2. Results

### 2.1. Endophytic Communities in the Seeds (Illumina Sequencing and Alignment)

In all plant seeds tested, there was a strong presence of bacteria from the phyla Firmicutes and Proteobacteria. A few species, such as *A. thaliana* (thale cress), *B. vulgaris* subsp. *cicla* (rainbow chard), *C. pepo* subsp. *pepo* (squash), and *N. caerulescens* (penny cress), also had a small presence of Actinobacteriota (Figure 2). The seed endophytic community of all crop species tested reacted differently to the consortium and the isolate, with no notable trend except for the disappearance of phylum Actinobacteriota, in three out of the four cases where it was present in the non-inoculated reference.

Another pattern of note is that the presence of Proteobacteria diminishes in six out of eight cases after inoculation with the bacterial consortium. The community structure shown in the non-inoculated column of penny cress can be assumed as the make-up of the consortium inoculation.

Figure 3 denotes the changes at the class level following seed inoculation with either the consortium or isolate. In seven of the eight cases, class Bacilli increased in relative abundance. The presence of Actinobacteria seems to occur only in the original community structure of the crops tested, with the only exception being N. caerulescens (penny cress), where there is an increase in Actinobacteria following inoculation with the consortium.

In six of the eight cases, the presence of Gammaproteobacteria decreased quite drastically following inoculation with the consortium, whereas this class only decreased in four of the eight cases when the plants were inoculated with the isolate (Figure 3).

Species richness analysis with an ANOVA indicated significant differences (*p* < 0.05) between both test groups and the non-inoculate reference in the case of M. sativa (alfalfa), while no notable difference was found between consortium inoculation and isolate inoculation (Figure 4). In this case, diversity decreased with both inoculations. Interestingly, a similar effect was also seen in *Z. mays* (corn) where there was a significant difference between the non-inoculated reference and the consortium inoculation, but not between the non-inoculated control and the inoculated isolate. However, looking at *Z. mays* (corn) through the lens of the Shannon and Simpson diversity index, it becomes apparent that the non-inoculated reference and isolate inoculation are significantly different, with the non-inoculated reference community having more diversity.

When *N. caerulescens* (penny cress) was inoculated with its own endophytes, the diversity within the seed increased significantly. This is important because in all other inoculations with the N. caerulescens consortium, the seed endophytes led to decreases in diversity and species richness. *H. annus* (sunflower), *C. pepo* subsp. *pepo* (squash), and *B. vulgaris* subsp. *cicla* (chard) showed no significant differences regarding species richness and diversity following inoculation. With regard to evenness, in seven of the eight tested species, excluding *H. annus* (sunflower), a significant difference was found relative to the non-inoculated reference.

### 2.2. In Planta Tests Results for Germination and Plant Weight

*N. caerulescens*, *M. sativa*, and *A. thaliana* responded differently to the seed inoculations in terms of plant growth and germination.

#### 2.2.1. *N. caerulescens*

In the case of *N. caerulescens* (penny cress), there was a significant difference (*p* < 0.001) in the plant sizes of those inoculated with the consortium, versus both the non-inoculated reference and the inoculated isolate (Figure 5).

There was no evident effect on germination (Figure 6).

#### 2.2.2. *M. sativa*

Inoculation of *M. sativa* (alfalfa) seeds resulted in no significant increases in plant size; however, there was an observed increase following the consortium inoculation (Figure 7).

Despite there not being any significant differences in plant size, inoculation with the seed endophyte consortium was found to have a pronounced significant effect on germination (*p* < 0.05) (Figure 8).

#### 2.2.3. *A. thaliana*

Inoculation of *A. thaliana* (thale cress) seeds with *S. wittichii* led to a significant decrease (*p* < 0.05) in plant size relative to the non-inoculated reference. Inoculation with the endophytic consortium neither hindered nor improved plant growth (Figure 9).

There was no pronounced effect on germination (Figure 10).

## 3. Discussion

The Illumina MiSeq sequencing delivered some interesting outcomes in terms of highlighting the core seed microbiome of different non-inoculated plant species as well as showing how inoculating seeds of these species changes that microbiome.

### 3.1. A. thaliana (Thale Cress)

The original seed microbiome of *A. thaliana* was shown to possess higher levels of Actinobacteriota when compared to all other sequenced seed microbiomes. This is notable because after both inoculations, the Actinobacteriota were reduced to levels too small to be visible in terms of community relative abundance. Actinobacteriota, specifically those from class Actinobacteria (Figure 2), have been shown to be integral in activating key defensive genes within *A. thaliana* [17]. It also had the only notable presence of Bacteroidota. Comparing our results to previous studies performed on the seed microbiome of *A. thaliana* confirms that there is a domination of Firmicutes and Proteobacteria within the seed [18]. The proportionally large amount of Firmicutes present compared to the Proteobacteria seems to be a common trend found in *A. thaliana* [6,18], which means that despite its strong history as a model organism for plant science, in terms of seed endophyte science, the results it generates may not be directly comparable to other seeds.

When looking at Pielou’s evenness for thale cress, we can see a significant difference between the communities following each inoculation. The inoculations themselves also differ significantly from each other. Linking this back to what we see in Figure 2, inoculation with either an isolate bacterium or a consortium lowers the endophytic diversity, specifically by reducing the amounts of Actinobacteriota and Bacteroidota. This may have a negative impact on future plant health since the symbionts associated with those phyla are commonly linked to plant protection from fungal pathogens [17,19]. This loss of diversity is again supported by the Shannon and Simpson Indexes (Figure 4), where the number of communities in the plants grown from inoculated seeds is shown to be significantly fewer than that of the non-inoculated reference. When these insights were applied to the in planta test (Figure 8), we speculated that the diminishment in two prevalent phyla occurring in the non-inoculated reference, and the overabundance of Firmicutes led to the decrease in plant size. Still, despite the slight decrease in plant size, an observed increase in germination was noted (Figure 9). Results from Tyc et al. [20] indicate that bacteria from these aforementioned phyla can indeed enhance germination through the release of volatile organic compounds (VOCs).

### 3.2. M. sativa (Alfalfa)

The core microbiome associated with *M. sativa* has previously been found to possess a large amount of Actinobacteria [21]. However, this previous study did not investigate the relative abundance. When compared to ours, it becomes apparent that the dominant phylum within the seed is Proteobacteria, which accounts for slightly more than 50% of the present species (Figure 2). The amount of Proteobacteria increases drastically when inoculated with the consortium and diminishes after inoculation of *S. wittichii*, our bacterial isolate. *S. wittichii* falls under the phylum Proteobacteria, and therefore the decrease in this phylum is noteworthy. It is a common trend in our results that inoculation with this bacterial species leads to a decrease in Proteobacteria, with only a few seed microbiomes being the exception.

This bacterium, although revered for its ability to break down dioxins and its potential to promote plant growth [14,15,16], has also been noted to possess a high amount of virulence factors associated with it, including seven genes that encode for an xcp secretion system. This secretion system is commonly used to secrete toxins into the extracellular fluid [22]. The research by Saeb also indicated that this bacterium acquires its virulence through horizontal gene transfer. It is likely that the introduction of this bacterium into the seed microbiome caused some detriment to the local Proteobacteria, either through virulence or competition.

In terms of evenness and species richness, both inoculations led to a significant decrease when compared to the non-inoculated reference; however, the resulting microbiomes of each inoculation did not differ from each other.

*M. sativa* was the only plant from our in planta tests that resulted in a significant increase in germination. This increase resulted from the inoculation with the *N. caerulescens* seed endophyte consortium (Figure 7). We believe this germination increase to be linked to the strong increase in Proteobacteria. Certain Proteobacteria have been shown to support germination under stressful conditions [20,21,23,24]. It is possible that treating seeds with a consortium is a good way to introduce more Proteobacteria into the seed without causing detriment to the plant (Figure 6). Neither inoculation had an impact on plant size; however, this can be considered a good thing because germination increased and plant health was not harmed.

### 3.3. N. caerulescens (Penny Cress)

The core seed microbiome of *N. caerulescens* has been shown to harbor Proteobacteria as its dominant phylum [10,25], as well as a steady amount of Actinobacteriota. Interestingly, *N. caerulescens* was the only plant species to maintain its presence of Actinobacteriota following both inoculations (Figure 2). This could be because the bacteria being introduced to the seed microbiome were not foreign, since they originated from seeds of this plant species. When looking at Pielou’s evenness (Figure 4), there is only one significant difference between the consortium inoculation and the non-inoculated reference. Surprisingly, inoculation with the consortium increased evenness significantly, which did not occur in any other plant species in our study. Diversity also significantly increased when *N. caerulescens* was treated with its own seed microbiome. This is again unique to *N. caerulescens* alone in this study. When these results are combined with the results of the in planta tests, we also see a corresponding increase in plant size (Figure 5).

Based on these results, we suggest that an ideal way to increase plant size is to treat the seeds with a consortium of their own seed endophytes, but further research is needed in this regard.

*N. caerulescens* is also the only plant species treated to increase its level of Actinobacteria following inoculation with the consortium (Figure 3). Actinobacteria have been shown to have a strong variety of plant-growth-promoting agents, including but not limited to synthesizing bioactive metabolites [26], ACC-deaminase production, siderophore production, nitrogen-fixation properties, and IAA production [27]. It is likely that having an increase in these bacteria in the seed prior to germination encourages plant growth and health, especially in stressful environments.

### 3.4. H. annuus (Sunflower)

*H. annuus* was the only tested species to show a notable increase in Actinobacteriota when inoculated with *S. wittichii* (Figure 2). Based on what was observed with the *N. caerulescens* in planta experiment, we assume that inoculation with this isolate will lead to an increase in overall plant size. Levels of proteobacteria also increased following both inoculations, indicating that there might be an increase in germination potential. There were no significant differences found in the community alpha analysis, which is indicative of a similar level of diversity being maintained. Overall, *H. annus* has a strong potential for positive outcomes when treated with seed endophytes.

### 3.5. Z. mays, R. sativus, and C. pepo subsp. pepo (Corn, Radish, and Squash)

*Z. mays*, *R. sativus*, and *C. pepo* subsp. *pepo* all showed an overwhelming loss of Proteobacteria following both inoculations (Figure 2). Because of this, they are likely not great candidates for inoculation with *N. caerulescens* endophytes.

In terms of *Z. mays*, it has been found that the seeds have extremely different levels of seed endophytes depending on the age of the seed and the stage of seed development [28]. Because of this, and the results of this study, this crop might not be an ideal candidate for inoculation with endophytes. However, future applications should look at the potential of all these plants when treated with an endophytic consortium or isolate from their own microbiome.

### 3.6. B. vulgaris subsp. cicla (Rainbow Chard)

To our knowledge, this is the first study to look at the seed endophytes of *B. vulgaris* subsp. *cicla*. Like all other plants in this study, apart from *A. thaliana*, *B. vulgaris* subsp. *cicla* had a relatively high abundance of Proteobacteria in its initial seed microbiome. It also possessed notable levels of Firmicutes and Actinobacteriota. Seed inoculation with *S. wittichii* leads to an increase in Proteobacteria- but a diminishment in Actinobacteria, suggesting that this is a possible inoculation option to increase germination, but not plant size (Figure 2). Pielou’s evenness showed that inoculation with this bacterium leads to significant amplification in one phylum when compared to the resulting community of the consortium inoculation or the non-inoculated reference. Species diversity was not impacted at a significant level, meaning that this plant would likely respond well to inoculation with *N. caerulescens* seed endophytes.

## 4. Materials and Methods

### 4.1. Bacterial Isolation and Consortium Enrichment of Seed Endophytes from N. caerulescens

*N. caerulescens* seeds were collected from a heavy-metal-polluted site in Plombières, Belgium (50.7379° N, 5.9611° E). Bacterial isolation and identification were obtained following the methods described in Langill et al., 2023. *S. wittichii* was selected to test as a bacterial isolate because of its history as a plant growth promoter, as well as the results obtained in our previous study where it was the one isolate to have a positive impact on *A. thaliana* [10,14]. In order to prepare the endophyte consortium, 10 µL of unquantified seed solution (5 g *N. caerulescens* seed paste suspended in 200 µL d H_2_O) was added to 100 µL of 1/10-diluted enriched liquid media (0.070 g CaCl_2_·2H_2_O, 0.200 g D-(+)-glucose·H_2_O, 1 g NaCl, 2 g Tryptone, 1 g Yeast Extract) [29]. This solution was incubated at 150 rpm at 30 °C for 3 days. Following this, the solution was centrifuged at 2000× *g* RPM for 10 min and subsequently washed with Phosphate-Buffered Saline (PBS) solution [30]. The washing step was repeated, and the final bacterial pellet was resuspended in dH_2_O, in order to make our consortium solution that was used for the seed inoculations mentioned in this paper. This pellet was subsequently sequenced using 16S rDNA gene Sanger sequencing to offer insight into the bacterial makeup of the consortium and additionally published in a previous paper [10].

### 4.2. Seed Selection and Inoculation

Seeds of eight plant species were selected to undergo inoculation of both the bacterial isolate *S. wittichii* and the consortium of seed bacteria collected from *N. caerulescens*. Seeds were selected to have a large variety of coat thickness, size, and general morphology (Table 1).

Seeds were surface-sterilized with 70% ethanol for a duration of 1 min. The seeds were subsequently washed with dH_2_O, and the sterilization method was repeated. Seeds were then divided into 3 inoculation groups and submerged in the solutions accordingly: non-inoculated (sterile dH_2_O), single isolate inoculation (*S. wittichii*), and consortium inoculation. Seeds were left in the inoculum solution for 3 days, in the dark, at a temperature of 5 °C. Following this, seeds underwent DNA extraction and preparation for Illumina sequencing.

Because there was an abundance of seeds in the groups of *A. thaliana*, *N. caerulescens*, and *M. sativa*, additional seeds were taken to be sown on heavy-metal-polluted soil originating from Overpelt, Belgium, to offer preliminary insight into how different inoculations affect germination and plant size. The composition of this soil has previously been described in our paper from 2023 [10].

### 4.3. DNA Extraction and Sequencing

DNA extraction of the seeds was conducted in replicates of 10, including a negative control. All DNA extractions were performed using the BioECHO EchoLUTION Plant DNA kit (Product number:010-003-050, Köln, Germany). All DNA samples were subjected to bacterial 16S rRNA gene amplicon PCR. In the first round of 16S rRNA gene PCR, an amplicon of 444 bp was generated, using primers 341F and 785R [31], using an Illumina adapter overhang nucleotide sequence (underlined), resulting in the following sequences, 341F-adaptor: 5′-TCG TCG GCA GCG TCA GAT GTG TAT AAG AGA CAG CCT ACG GGN GGC WGC AG-3′ and 785R-adaptor: 5′-GTC TCG TGG GCT CGG AGA TGT GTA TAA GAG ACA GGA CTA CHV GGG TAT CTA ATC C-3′. Using the Q5 High-Fidelity DNA Polymerase system (M0491, NEB), a reaction volume of 25 μL per sample was prepared containing 1 μL of extracted DNA (final DNA concentration per reaction, 1–10 ng), 1× Q5 Reaction Buffer with 2 mM MgCl2, 200 μM dNTP mix, 1× Q5 High GC Enhancer (for the soil and fungi samples), 0.2 μM forward or reverse primer, and 1.2 U Q5 High-Fidelity DNA polymerase. For the seed endophytic samples, 1 μL mitoPNA blocker (2 μM final concentration added from a 50 μM stock) and 1 μL plastidPNA blocker (2 μM final concentration from 50 μM stock) were additionally prepared. The PCR program started with an initial denaturation for 3 min at 98 °C, followed by a 30 s denaturation at 98 °C, a 30 s annealing at 57 °C for V3V4 (58 °C for ITS), and a 1 min extension at 72 °C; all three steps were repeated for a total of 35 cycles. The reaction was ended by a final 7 min extension at 72 °C. The amplified DNA was purified using the AMPure XP beads (Beckman Coulter, Brea, CA, USA) and the MagMax magnetic particle processor (ThermoFisher, Leuven, Belgium). Subsequently, 5 μL of the cleaned PCR product was used for the second PCR attaching the Nextera indices (Nextera XT Index Kit v2 Set A (FC-131-2001), and D (FC-131-2004), Illumina, Antwerp, Belgium). For these PCR reactions, 5 μL of the purified PCR product was used in a 25 μL reaction volume and prepared following the 16S Metagenomic Sequencing Library Preparation Guide. PCR conditions were the same as described above, but the number of cycles was reduced to annealing temperatures of 20 and 55 °C. PCR products were cleaned using the Agencourt AMPure XP kit, and then quantified using the Qubit dsDNA HS assay kit (Invitrogen, Waltham, MA, USA) and the Qubit 2.0 Fluorometer (Invitrogen). Once the molarity of the sample was determined, the samples were diluted down to 4 nM using 10 mM Tris pH 8.5 prior to sequencing on the Illumina MiSeq. Samples were sequenced using the MiSeq Reagent Kit v3 (600 cycles) (MS-102-3003) and 15% PhiX Control v3 (FC-110-3001). For quality control, a DNA-extraction blank and PCR blank were included throughout the process, and the ZymoBIOMICS Microbial Mock Community Standard (D6300) was used to test the efficiency of DNA extraction (Zymo Research, Irvine, CA, USA).

### 4.4. Data Visualization and Statistical Analyses

The ASV table was further processed by removing chloroplasts and mitochondria, and prevalence-filtered using a 2% inclusion threshold (unsupervised filtering) as described by Callahan et al., 2016 [32]. Alpha-diversity metrics such as observed ASV count, and Simpson’s and Shannon’s diversity indexes were calculated on unfiltered data using scripts from the MicrobiomeSeq package for R version 4.2.1. Hypothesis testing was performed using analysis of variance (ANOVA) and the Tukey Honest Significant Differences method (Tukey HSD). When assumptions of normality and homoscedasticity were not met, a Kruskal–Wallis Rank Sum test and a Wilcoxon Rank Sum test were performed. The results were summarized in boxplots. For beta-diversity, the Bray–Curtis, weighted, and unweighted UniFrac distances were calculated on unfiltered data using the vegan package (version 2.5.4), and the data were visualized using a non-metric multidimensional scaling (NMDS). Relative abundances were calculated and visualized in bar charts using Phyloseq [33]. All graphs were generated in R version 4.2.1 and SPSS 28.0.1.0. Additionally, a chi-squared analysis was performed to quantify the presence of a relationship between germination and inoculation. Bar charts with information pertaining to germination were created in SPSS 28.0.1.0.

### 4.5. Sowing, Germination, and Harvest

Seeds from the aforementioned plant species of *A.thaliana*, *M. sativa*, and *N. caerulescens* were sown on polluted soil from Overpelt, Belgium [10]. Overpelt is known to have high Zn, Cd, and Pb pollution in the soil, making it an ideal marginal soil to stress the seeds [10]. For each plant species and inoculation (no inoculation, consortium, *S. wittichii*), 40 seeds were planted in order to ensure enough replicates would be available for further analysis. Plants were grown in the greenhouse (170 mol m^−2^ s^−1^ PAR, 22 °C/18 °C degrees, 12/12 day–night cycle, and 65% RH). For germination, the first 20 seeds planted were monitored and germination status was recorded after 5 days.

Plants were harvested after 3 weeks, and all aboveground aspects of the plant were weighed. This is recorded as fresh plant weight. Roots were not included in this study.

## 5. Conclusions

The inoculation of seeds of different plant species with the seed endophytes of *N. caerulescens* and with the single bacterial isolate *S. wittichii* led to significant community changes within the ungerminated seed. This occurred regardless of plant species. Some community changes had the potential to be beneficial based on the plant growth tests conducted in this study, but for some species, such as *Z. mays*, *R. sativus*, and *C. pepo* subsp. *pepo*, a different avenue of seed inoculation should be taken in order to increase germination and plant growth under stressful conditions. Crop species similar to *M. sativa* or *B. vulgaris* subsp. *cicla* have the potential to respond very positively to this inoculation. We noticed a significant increase in the plant size of *N. caerulescens* when it was inoculated with its own seed endophytic consortium. This could be linked to the increase in Actinobacteria found in the seed microbiome post inoculation. *M. sativa* showed a significant increase in germination potential after inoculation with the seed endophyte consortium. Community structure changes showed a dramatic increase in Proteobacteria. *A. thaliana* had a notable decrease in plant size following inoculation with bacterial isolate *S. wittichii*, but *A. thaliana* also had a notable difference in community make-up when compared to the rest of the plants observed in this study. We propose that in order to maximize seed germination and plant size of any plant species, further studies should address the impact of a local amplification of a species’ own seed endophyte, rather than focusing on the introduction of potentially foreign endophytes.

## Figures and Tables

**Figure 1 plants-12-03660-f001:**
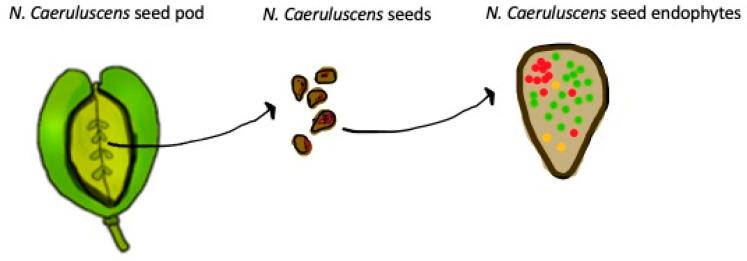
*N. caerulescens* seed pod, associated seeds, and internal seed endophytes. The seed endophytes are pictured as coloured dots.

**Figure 2 plants-12-03660-f002:**
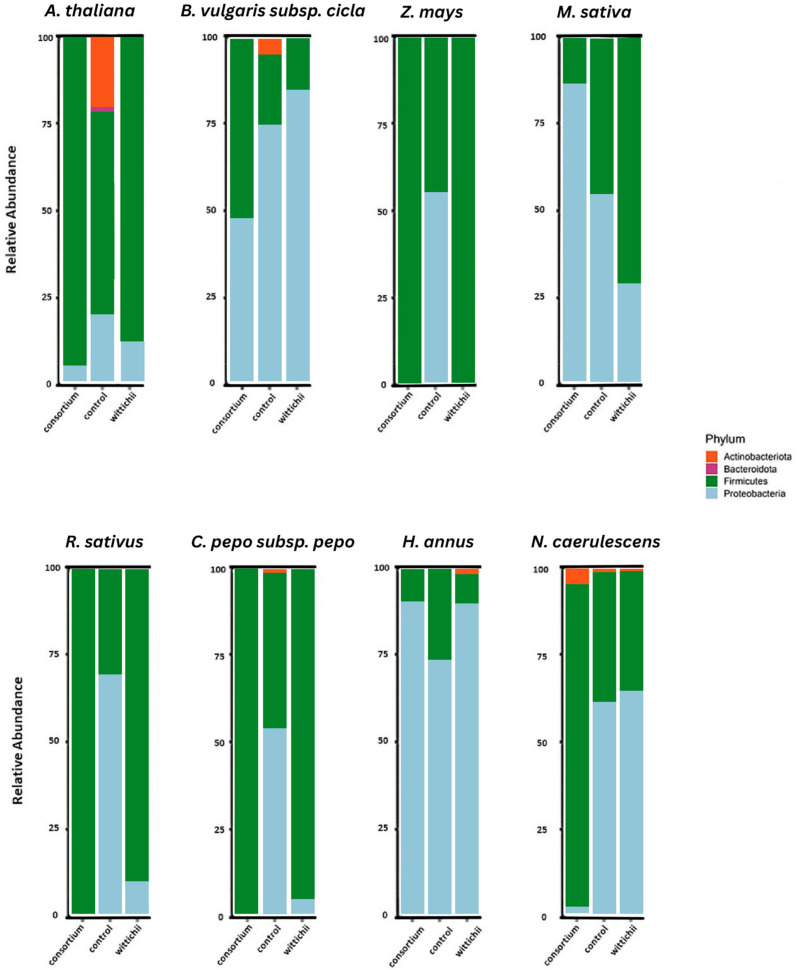
Relative abundance of the phyla that make up the bacterial endophytic community within each corresponding plant. The community structure changes are shown in each bar chart with consortium inoculation to the left of the non-inoculated reference, and isolate inoculation to the right. The community make-up of non-inoculated Penny cress is the likely make-up of the consortium inoculation used for inoculation in this paper.

**Figure 3 plants-12-03660-f003:**
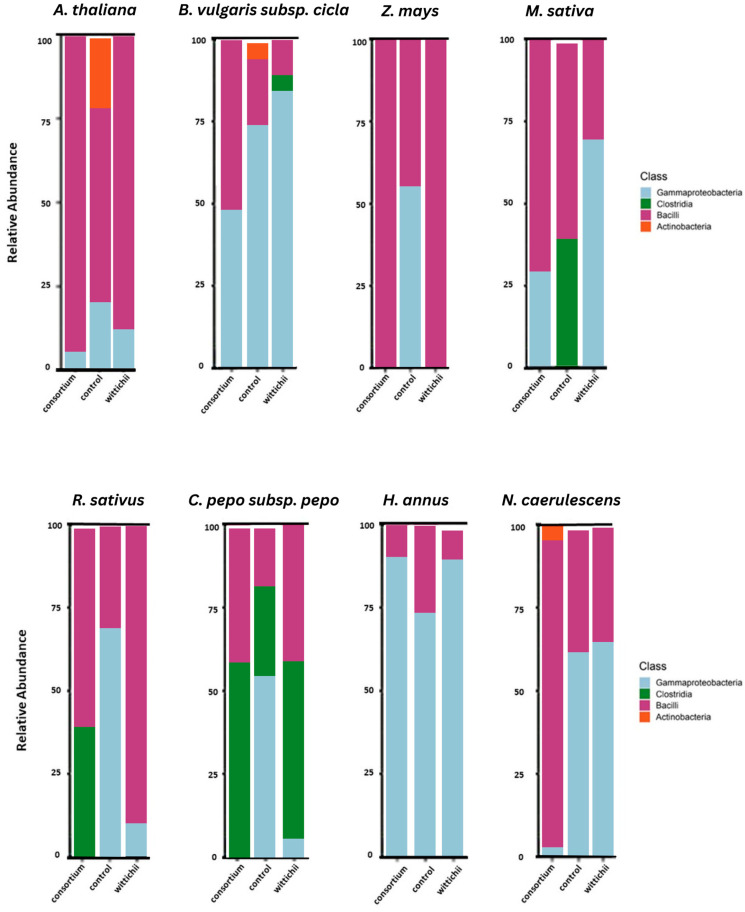
Relative abundance of classes of bacteria that make up the seed endophyte community within each corresponding plant species. The community structure changes are shown in each bar chart with consortium inoculation to the left of the non-inoculated reference, and isolate inoculation to the right.

**Figure 4 plants-12-03660-f004:**
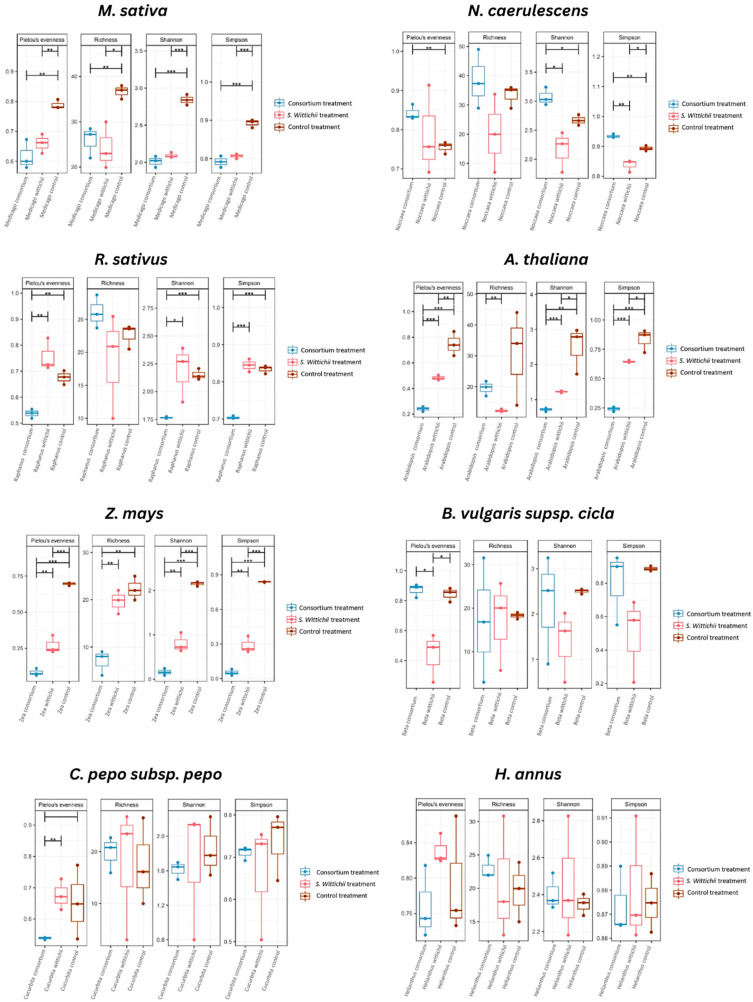
The species richness and diversity indexes of the endophytic communities of the tested plant species, based on inoculation with the seed endophytic consortium from *N. caerulescens*, and inoculation with bacterial isolate *S. wittichii*. A significance of *p* < 0.05 is indicated with a single asterix (*), while a significance of *p* < 0.01 is shown with a double asterix (**), and a significance of *p* < 0.001 is indicated with a triple asterix (***).

**Figure 5 plants-12-03660-f005:**
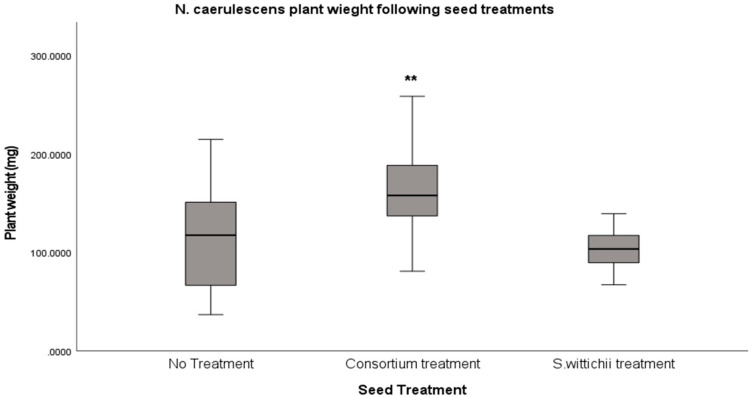
Average fresh weight (mg) of *N. caerulescens* following inoculation with seed endophyte consortium, and of the single bacterial isolate *S. wittichii.* An ANOVA with Tukey HSD test was performed, and a significant difference (*p* < 0.001) was found between the consortium inoculations when compared to both the non-inoculated reference and the inoculated *S. wittichii*. This significance is highlighted with a double asterisk in the figure (**).

**Figure 6 plants-12-03660-f006:**
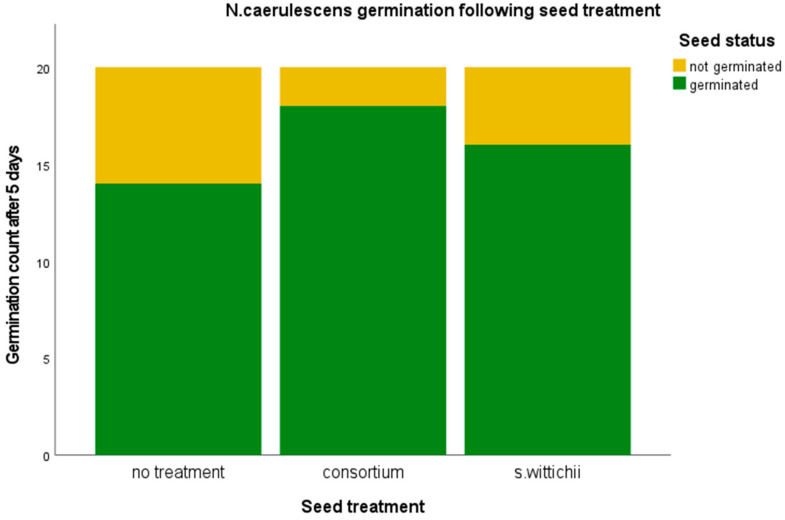
Germination percentage of *N. caerulescens* following inoculation with the seed endophyte consortium, and of the single bacterial isolate *S. wittichii.* A chi-squared test was performed. No significant relationships were found.

**Figure 7 plants-12-03660-f007:**
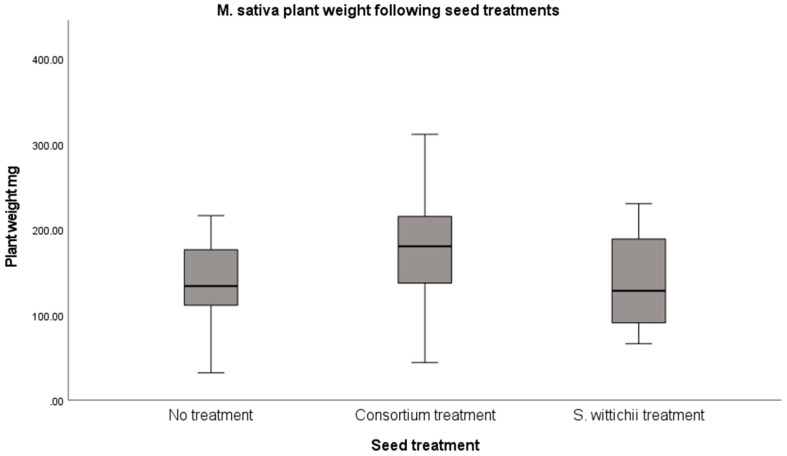
Average fresh weight of *M. sativa* following inoculation with the seed endophyte consortium, and of the single bacterial isolate S. wittichii. An ANOVA with Tukey’s HSD test was performed. No significant differences were found.

**Figure 8 plants-12-03660-f008:**
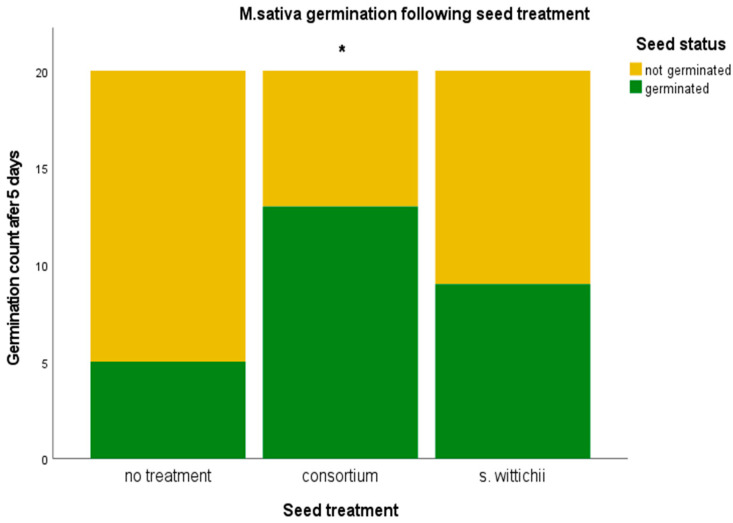
Germination percentage of *M. sativa* following inoculation with the seed endophyte consortium, and of the single bacterial isolate *S. wittichii.* A chi-squared test was performed and showed a significant relationship between the seed inoculation with the bacterial consortium and increased germination under stressful conditions. A significance of *p* < 0.05 is indicated by an asterix (*).

**Figure 9 plants-12-03660-f009:**
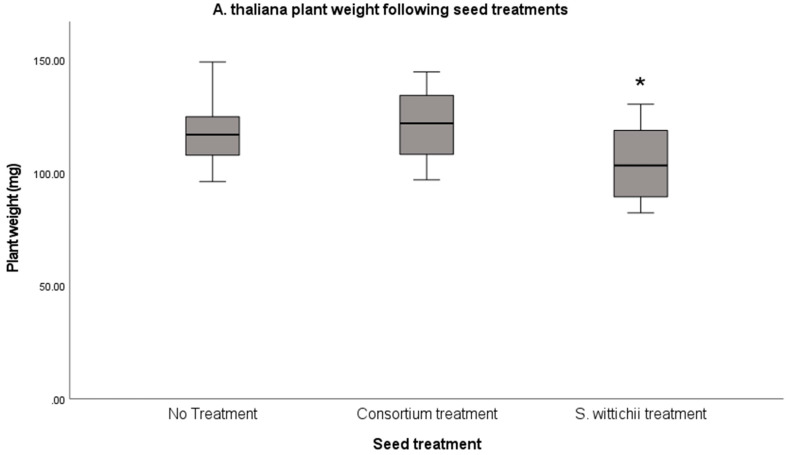
Average fresh weight of *A. thaliana* following inoculation with the seed endophyte consortium, and of the single bacterial isolate S. wittichii. An ANOVA with Tukey’s HSD test was performed showing a significant (*p* < 0.05) and smaller plant size relative to the non-inoculated reference. This significance is highlighted with an asterix (*).

**Figure 10 plants-12-03660-f010:**
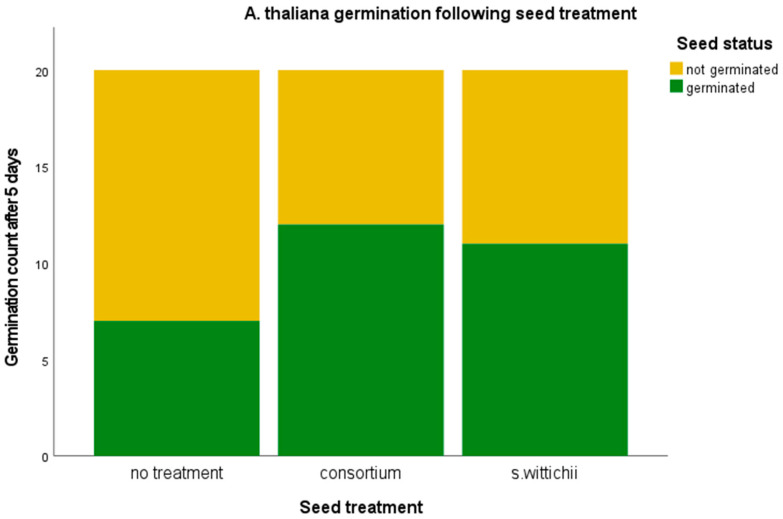
Germination percentage of *A. thaliana* following inoculation with the seed endophyte consortium, and of the single bacterial isolate *S. wittichii.* A chi-squared test was performed. No significant relationships were found.

**Table 1 plants-12-03660-t001:** An overview of the scientific name, the common name, and the images of the seeds used in the isolate and consortium inoculation.

Scientific Name	Common Name	Seed Image	Seed Size
*Medicago sativa*	Alfalfa	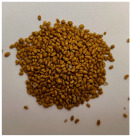	1–2 mm
*Arabidopsis thaliana*	Thale cress	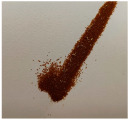	0.3–0.5mm
*Zea mays*	Corn	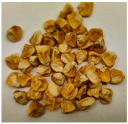	6–9 mm
*Raphanus sativus*	Radish	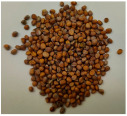	1–3 mm
*Noccaea caerulescens*	Penny cress	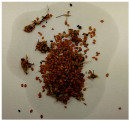	0.6–0.8 mm
*Helianthus annus*	Sunflower	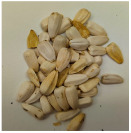	6–10 mm
*Cucurbita pepo* subsp. *pepo*	Squash	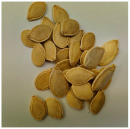	8–14 mm
*Beta vulgaris* subsp. *cicla*	Rainbow chard	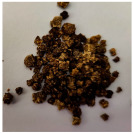	1–2 mm

## Data Availability

BioSample metadata are available in the NCBI Biosample database (http://ncbi.nlm.nih.gov/biosample/ (accessed on 15 October 2023)) under accession numbers: SAMN37890229-SAMN37890302.

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
