# Peer review of "Endophyte Community Changes in the Seeds of Eight Plant Species following Inoculation with a Multi-Endophytic Bacterial Consortium and an Individual Sphingomonas wittichii Strain Obtained from Noccaea caerulescens"

_plants, 2023, doi:10.3390/plants12203660_

Round 1
Reviewer 1 Report
1. In the abstract part, the description about results is too easy which cannot make readers understand the promising results;
2. The species names of plants should have scientific name in the abstract;
3. I don’t think “Pb-Zn hyperaccumulator” is one of keywords;
4. Figure 1 and Table 1 did not provide enough information which can be put into supplementary material. Or Table 1 should provide more information like seed size range. And also, the seed images should have scale;
5. Figure 5 should explain the meaning of “**”;
Author Response
Dear reviewer,
Thank you for taking the time to read our manuscript. Your review has proved valuable as it points out some areas for improvement. I hope we can agree with the changes I have made to the document. I have highlighted the changes specific to your review in yellow.
- In the abstract part, the description about results is too easy which cannot make readers understand the promising results;
- I am unsure what actions you would like to see me take to rectify this
- The species names of plants should have scientific name in the abstract;
- Agreed. This has now been changed in the abstract
- I don’t think “Pb-Zn hyperaccumulator” is one of keywords;
- I have changed the keyword to hyperaccumulator and adjusted the order accordingly
- Figure 1 and Table 1 did not provide enough information which can be put into supplementary material. Or Table 1 should provide more information like seed size range. And also, the seed images should have scale;
- I have added the seed size range to Table 1.
- Figure 5 should explain the meaning of “**”;
- I have added additional explanation of the double asterix.
Reviewer 2 Report
The paper describes effect of bacterial consortium on seed endophytic microbiome. The purpose of taking up this research is not very clear, while the objectives are also not well defined. The concerns are given below:
1. Title need to be revised. "Community changes in 8 plant species" is incorrect as the microbiome of whole plant has not been studied here (or is it? Because its not given in methods). Also, different plant parts have different communities with their own dynamics.
2. Need to justify, why seeds of these eight crops were treated with a consortium of bacteria prepared from isolates of Noccaea caerulescens, but NOT with a consortium derived form the respective host plant itself.
3. What was the logic of making this consortium? Does this consortium has any beneficial effects on the metal accumulation activity? In fact, the composition of consortium has not been described. It remain a mystery as all the culturable bacteria, with variable growth rates had been applied. Not sure if the such composition would remain consistent in terms of composition and quantity.
4. The hypothesis given in line 92-94 does not stand to its merit, if really this can be taken forward for commercialization.
5. Its not given how endophytic communities has been characterized after consortium treatment. Section 4.3 directly starts from DNA extraction.
6. What is seed solution. (Section 4.1)
7. The variation in phyla and classes of bacteria due to the treatments should have some significance, else its a known phenomenon the microbial communities in plants' tissues are highly dynamic, and varies with age, growth conditions, environmental changes, biotic and abiotic factors. How these have been taken into consideration?
Author Response
Dear reviewer,
The time and effort you have spend reviewing this is appreciated. I hope we can agree on the changes I have made to the manuscript in order to accommodate your concerns, as well as those of your peer reviewers. Please find my answers to your comments below. All changes made specifically for you in the manuscript are highlighted in green.
The paper describes effect of bacterial consortium on seed endophytic microbiome. The purpose of taking up this research is not very clear, while the objectives are also not well defined. The concerns are given below:
- Title need to be revised. "Community changes in 8 plant species" is incorrect as the microbiome of whole plant has not been studied here (or is it? Because its not given in methods). Also, different plant parts have different communities with their own dynamics.
- The title of the paper is "Seed endophyte community changes...". This paper pertains only to the bacterial endophytic community of seeds.
2. Need to justify, why seeds of these eight crops were treated with a consortium of bacteria prepared from isolates of Noccaea caerulescens, but NOT with a consortium derived form the respective host plant itself.
- This point is indeed a great idea - it is already being undertaken as a follow-up project to the one that was completed in this paper. Here, we look specifically at N. caerulescens because it is a hyperaccumulator and we believed that the seed endophytes would be more helpful for encouraging growth on marginal land than simply treating the plants with an enriched consortium from their own seed endophytes. However, the results of this paper support the idea that this is a direction we should proceed in, moving forward.
3. What was the logic of making this consortium? Does this consortium has any beneficial effects on the metal accumulation activity? In fact, the composition of consortium has not been described. It remain a mystery as all the culturable bacteria, with variable growth rates had been applied. Not sure if the such composition would remain consistent in terms of composition and quantity.
- The logic behind making a bacterial consortium is summarized in lines 92-94. The composition of the consortium is described in lines 113-115.
4. The hypothesis given in line 92-94 does not stand to its merit, if really this can be taken forward for commercialization.
- This hypothesis has now been reworded to make it more applicable to the proposed publication.
5. Its not given how endophytic communities has been characterized after consortium treatment. Section 4.3 directly starts from DNA extraction.
- The communities were characterized with illumina sequencing. Additionally information has been added to the manuscript to highlight what was done.
6. What is seed solution. (Section 4.1)
- Added description
7. The variation in phyla and classes of bacteria due to the treatments should have some significance, else its a known phenomenon the microbial communities in plants' tissues are highly dynamic, and varies with age, growth conditions, environmental changes, biotic and abiotic factors. How these have been taken into consideration?
- This was taken into account by using a control to which the treatments were compared. The control underwent all the same treatments sans a bacterial inoculum, which should indicate that the base endophytic community is what would have existed should the treatments not have been performed.
Reviewer 3 Report
The subject of the research and the conducted experiment is consistent with the theme of Plants journal. The results of this research are of great practical importance. The work is extensive and contains many results of practical and important importance for scientific research. However, I believe that the topic of the thesis is too long and I propose to shorten the topic of the thesis. All drawings in the work are difficult to read, it is recommended to number them, e.g.: A, B, C.....
Summary: contains information about the experiment performed, but there are very few research results. Too much introductory content. Please enter where and in what years the experiment was conducted.
Introduction:
The introduction presents the issues of the work. The authors included the purpose of the work.
Results
All drawings require improvement in terms of readability.
Discussion
Discussion and it raises no objections.
Materials and methods
There are no soil test results for metal content. The year of the experiment is also missing.
Conclusions
Results too general, no specific results.
Literature
Current, well-selected literature.
After proofreading, the work may be published in the Plants journal.
Author Response
Dear reviewer,
Thank you for taking the time to assess the quality of our manuscript. Your insight is invaluable in helping us provide the best product possible. I hope we can agree on the changes I've made to the manuscript in order to accomadate your review.
The subject of the research and the conducted experiment is consistent with the theme of Plants journal. The results of this research are of great practical importance. The work is extensive and contains many results of practical and important importance for scientific research. However, I believe that the topic of the thesis is too long and I propose to shorten the topic of the thesis. All drawings in the work are difficult to read, it is recommended to number them, e.g.: A, B, C.....
Due to conflicting reviews on the length of the manuscript, I have opted to add in information where it is deemed necessary and try to keep all descriptions as concise as possible.
Summary: contains information about the experiment performed, but there are very few research results. Too much introductory content. Please enter where and in what years the experiment was conducted.
The experiment was performed in Belgium, with seeds originating from Plombieres Belgium. The experiment took place in 2022. The first published paper with results from this ongoing project was published in February 2023. This information has been added to the text
Introduction:
The introduction presents the issues of the work. The authors included the purpose of the work.
Results
All drawings require improvement in terms of readability.
I have increased the image size and clarity for better readability
Materials and methods
There are no soil test results for metal content. The year of the experiment is also missing.
I have added a line showing where that information can be found (highlighted in blue)
Conclusions
Results too general, no specific results.
Because this paper is exploratory, the results of it are quite general, with more follow-up experiments needed. It is my strong hope that this publication will be a foundation for more specific experiments and test in the future.
Round 2
Reviewer 2 Report
Strongly advise that the title should be reconsidered. May be - Endophyte community changes in seeds of different other plants following inoculation with a multispecies -consortium, or an individual Sphingomonas wittichii strain obtained from Noccaea caerulescens
Author Response
Dear reviewer,
I am happy to change the title. Perhaps we could agree on the following?
Endophyte community changes in the seeds of 8 plant species following inoculation with a multi-endophytic bacterial consortium and an individual Sphingomonas wittichii strain obtained from Noccaea caerulescens
Reviewer 3 Report
The authors of the work improved the manuscript in accordance with the reviewer. The work may be published in the journal Plants
Author Response
Dear reviewer,
Thank you for your assessment of our work!